# The SPARC complex defines RNAPII promoters in *Trypanosoma brucei*

**Desislava P Staneva[1,2], Stefan Bresson[1], Tatsiana Auchynnikava[1†], Christos Spanos[1], Juri Rappsilber[1,3], A Arockia Jeyaprakash[1], David Tollervey[1], Keith R Matthews[2]\*, Robin C Allshire[1]\***

[1]Wellcome Centre for Cell Biology, University of Edinburgh, Edinburgh, United Kingdom; [2]Institute of Immunology and Infection Biology, School of Biological Sciences, University of Edinburgh, Edinburgh, United Kingdom; [3]Institute of Biotechnology, Technische Universität, Berlin, Germany

**Abstract** Kinetoplastids are a highly divergent lineage of eukaryotes with unusual mechanisms for regulating gene expression. We previously surveyed 65 putative chromatin factors in the kinetoplastid *Trypanosoma brucei*. Our analyses revealed that the predicted histone methyltransferase SET27 and the Chromodomain protein CRD1 are tightly concentrated at RNAPII transcription start regions (TSRs). Here, we report that SET27 and CRD1, together with four previously uncharacterized constituents, form the SET27 promoter-associated regulatory complex (SPARC), which is specifically enriched at TSRs. SET27 loss leads to aberrant RNAPII recruitment to promoter sites, accumulation of polyadenylated transcripts upstream of normal transcription start sites, and conversion of some normally unidirectional promoters to bidirectional promoters. Transcriptome analysis in the absence of SET27 revealed upregulated mRNA expression in the vicinity of SPARC peaks within the main body of chromosomes in addition to derepression of genes encoding variant surface glycoproteins (VSGs) located in subtelomeric regions. These analyses uncover a novel chromatin-associated complex required to establish accurate promoter position and directionality.

**\*For correspondence:**
keith.matthews@ed.ac.uk (KRM);
robin.allshire@ed.ac.uk (RCA)

**Present address:** †The Francis Crick Institute, London, United Kingdom

**Competing interest:** The authors declare that no competing interests exist.

## Editor's evaluation

Trypanosomes are unusual in the way that they transcribe protein-coding genes. This work defines the role of the SPARC complex in transcription and highlights the role of potential histone readers and writers. This work will be of interest to those in the kinetoplastid community, especially those working on the control of gene expression.

## Introduction

Kinetoplastids are a group of highly divergent eukaryotes that separated from the main lineage early in eukaryotic evolution. Reflecting this, kinetoplastids display many unusual molecular and biochemical features distinct from those of conventional eukaryotes. For example, gene expression regulation in these organisms is very different from conventional model eukaryotes such as yeasts, fruit flies, nematodes, and vertebrates, all of which are members of the Opisthokonta clade (*Akiyoshi and Gull, 2013*; *Clayton, 2019*; *Keeling and Burki, 2019*).

*Trypanosoma brucei* is a diploid kinetoplastid parasite transmitted to mammals by the tsetse fly in sub-Saharan Africa, where it causes human sleeping sickness and animal nagana. In the mammalian host, bloodstream form (BF) trypanosomes have a surface coat composed of variant surface glycoprotein (VSG). Only one VSG is expressed at a time from a collection of ~2000 distinct VSG genes and gene fragments, many of which are clustered near telomeres (*Horn, 2014*). *T. brucei* evades the

mammalian immune system by occasionally switching to express a new VSG to which the host has not produced antibodies. Tsetse flies take up trypanosomes when they feed on infected mammals. In the fly midgut, *T. brucei* differentiates to the procyclic form (PF), a transition coupled to metabolic reprogramming and replacement of VSG with procyclins on the surface of these parasites (*Matthews, 2005*; *Smith et al., 2017*). Unusually, the active VSG and procyclin mRNAs are transcribed by RNAPI which is hypothesized to result from the high demand for these surface coat proteins at different stages of the *T. brucei* life cycle.

Most other *T. brucei* protein-coding genes are transcribed by RNAPII within long polycistrons from which individual mRNAs are processed using signals in adjacent 5′ and 3′ untranslated regions (UTRs; *Benz et al., 2005*; *Kolev et al., 2010*). Short spliced leader (SL) sequences are trans-spliced onto all pre-mRNAs, and this is mechanistically linked to the 3′ end processing of upstream genes (*Matthews et al., 1994*; *Michaeli, 2011*). Poly(A) tails are added to the 3′ end of all mRNAs, including VSG gene transcripts. Because of their polycistronic arrangement, RNAPII transcription of *T. brucei* genes is initiated at only ~150 sites in a genome that encodes ~9000 proteins (*Berriman et al., 2005*; *Daniels et al., 2010*). Such transcriptional start regions (TSRs) are analogous to the promoters of conventional eukaryotes and similarly exhibit bidirectional/divergent (dTSR) or unidirectional/single (sTSR) activity. TSRs are broad 5–10 kb regions marked by the presence of the histone variants H2A.Z and H2B.V and enriched for specific histone modifications, including methylation on H3K4 and H3K10, and acetylation on H3K23, H4A1, H4K2, H4K5, and H4K10 (*Kraus et al., 2020*; *Siegel et al., 2009*; *Wright et al., 2010*).

RNAPII transcription termination regions (TTRs) are marked by the presence of chromatin containing the H3.V and H4.V histone variants and the kinetoplastid-specific DNA modification base J (*Schulz et al., 2016*; *Siegel et al., 2009*). Approximately 60% of TTRs are located in regions where two neighboring polycistrons end (convergent TTRs, cTTRs). The remaining 40% of TTRs are found between polycistrons oriented head-to-tail (single TTRs, sTTRs), and frequently coincide with RNAPI or RNAPIII transcribed genes.

Regulation of trypanosome gene expression is thought to occur predominantly post-transcriptionally via control of pre-mRNA processing, turnover, and translation, suggesting a lack of promoter-mediated modulation. It was therefore surprising that a large number of putative chromatin regulatory factors exhibit enrichment in the vicinity of TSRs (*Schulz et al., 2015*; *Siegel et al., 2009*; *Staneva et al., 2021*). We previously identified two classes of TSR-associated factors. In chromatin immunoprecipitation assays, Class I factors, such as the predicted lysine methyltransferase SET27 and the Chromodomain protein CRD1, exhibit sharp peaks that coincide with the 5′ end of nascent pre-mRNAs. In contrast, Class II factors are distributed more broadly from the Class I peak and extend into downstream transcription units (*Staneva et al., 2021*).

Here, we set out to characterize the Class I factors SET27 and CRD1 which exhibit some of the most prominent peaks at RNAPII TSRs. Our previous affinity purifications and mass spectrometry identified four uncharacterized proteins that interact strongly with both SET27 and CRD1. We show that all six proteins associate with each other and that the four uncharacterized proteins also display sharp peaks at TSRs, coincident with sites where SET27 and CRD1 reside. Thus, these six proteins form the SET27 promoter-associated regulatory complex (SPARC). We further show that SPARC is required for accurate transcription initiation and/or promoter definition.

## Results

### SET27 and CRD1 associate with proteins containing chromatin reader modules

We previously reported that YFP-tagged SET27 (Tb927.9.13470) and CRD1 (Tb927.7.4540) co-immunoprecipitate with each other and with JBP2, a thymidine hydroxylase involved in the synthesis of base J (*DiPaolo et al., 2005*; *Kieft et al., 2007*). In addition, both SET27 and CRD1 interacted with four uncharacterized proteins (Tb927.11.11840, Tb927.3.2350, Tb927.1.4250, and Tb927.11.13820; *Staneva et al., 2021*). SET27 is predicted to contain a SET methyltransferase domain (*Dillon et al., 2005*) and a Zinc finger domain (*Klug and Rhodes, 1987*), whereas CRD1 has a putative methyl lysine binding Chromo domain (*Bannister et al., 2001*; *Lachner et al., 2001*; *Paro, 1990*; *Singh et al., 1991*; *Figure 1—figure supplement 1A*). Among the four uncharacterized proteins, Tb927.11.11840 was

named CSD1 because it contains a divergent Chromoshadow domain (*Aasland and Stewart, 1995*), normally found in association with Chromodomains. The uncharacterized protein Tb927.3.2350 has a predicted PHD finger histone methylation reader domain (*Aasland et al., 1995*; *Li et al., 2006*; *Peña et al., 2006*), and was thus designated as PHD6. Tb927.1.4250 lacked strongly predicted domains but because it was enriched at promoters (see below), we named it Promoter Binding Protein 1 (PBP1). Finally, Tb927.11.13820 was designated PWWP1 due to its structural similarity to a PWWP domain of the human NSD2 histone methyltransferase, which binds methylated H3K36 (*Arrowsmith and Schapira, 2019*; *Zhang et al., 2021*). Overall, the presence of putative histone writer and reader domains suggests that these proteins may cooperate to regulate chromatin structure and gene expression.

## Identification of the TSR-associated SPARC complex in BF *T. brucei*

The enrichment of CSD1, PHD6, PBP1, PWWP1, and JBP2 with affinity-selected YFP-SET27 and YFP-CRD1 (*Staneva et al., 2021*) initially suggested that these seven proteins might form a single complex. To investigate this further, we YFP-tagged the four uncharacterized proteins and JBP2 at their endogenous gene loci in BF Lister 427 *T. brucei* cells. We tagged the proteins on their N termini to preserve 3′ UTR sequences involved in regulating mRNA stability (*Clayton, 2019*). We note, however, that the presence of the YFP tag and/or its position (N- or C-terminal) might affect protein expression and localization patterns. Immunolocalization was performed to determine the subcellular distribution of the putative SET27-CRD1 complex components. Strong nuclear signals were obtained for CRD1, CSD1, PHD6, PBP1, and PWWP1, whereas SET27 and JBP2 were found in both the nucleus and the cytoplasm of BF cells (*Figure 1—figure supplement 1B*). We subsequently affinity purified each protein and identified its interacting partners via the same LC-MS/MS proteomics pipeline we previously used for YFP-SET27 and YFP-CRD1 (*Staneva et al., 2021*). This analysis showed that each protein displayed strong association with the other six (*Figure 1A*; *Supplementary file 1*).

This group of seven represented the most highly enriched proteins in the affinity selections of SET27, CRD1, CSD1, and PBP1, being on average eightfold more prevalent than any other proteins detected. However, several bait proteins recovered overlapping sets of additional proteins (*Figure 1B*; *Supplementary file 2*). SET27, CRD1, CSD1, PBP1, and PWWP1 associated with JBP3, a base J binding protein involved in RNAPII transcription termination in *T. brucei* and *Leishmania* (*Jensen et al., 2021*; *Kieft et al., 2020*). PHD6 and PWWP1 affinity purifications were enriched for various histones (H2A, H2B, H3.V, H4, etc.) and kinetochore proteins (KKT2, KKT4, KKT8, etc.), and PHD6, PBP1, and PWWP1 each associated with the histone variant H4.V and the kinetochore protein KKT3. Recovery of histone proteins is consistent with the presence of putative histone-binding modules in CRD1, PHD6, and PWWP1. Additionally, CRD1, PHD6, and PWWP1 interacted with various RNAPII subunits, and both PHD6 and PWWP1 associated with the POB3 and SPT16 components of the FACT complex. FACT is involved in transcription elongation in eukaryotes (*Belotserkovskaya et al., 2003*), and in *T. brucei* it has also been linked to VSG repression (*Denninger and Rudenko, 2014*). Overall, our proteomic analyses indicate that SET27, CRD1, CSD1, PHD6, PBP1, PWWP1, and JBP2 are tightly associated, potentially forming a protein complex, with some components showing interactions with a wider group of proteins involved in transcriptional regulation (*Figure 1B*; *Supplementary file 2*).

Previously we demonstrated that both SET27 and CRD1 are specifically enriched across a narrow segment of RNAPII TSRs in *T. brucei* BF cells (*Staneva et al., 2021*). We therefore performed ChIP-seq for the five other SET27/CRD1-associated proteins in BF parasites. CSD1, PHD6, PBP1, and PWWP1 each showed sharp peaks at TSRs, coincident with SET27 and CRD1 (*Figure 2A, B*). In contrast, we did not detect specific association of JBP2 with any genomic region. Thus, JBP2 is either not chromatin-associated, only transiently associated with chromatin or the fixation conditions used might be insufficient to crosslink it to chromatin. Consequently, despite the clear and robust affinity purification of JBP2 with the other six proteins in BF cells, we are unable to reliably designate JBP2 as a core component of the complex. Because SET27, CRD1, CSD1, PHD6, PBP1, and PWWP1 clearly associate with each other and are co-enriched over RNAPII promoter regions, we termed this six-membered complex the SET27 promoter-associated regulatory complex (SPARC).

## PF *T. brucei* cells also contain SPARC

To determine if SPARC exists with a similar composition in the distinct insect stage of the *T. brucei* life cycle, we YFP-tagged SET27, CRD1, and JBP2 at their endogenous loci in PF parasites. In PF

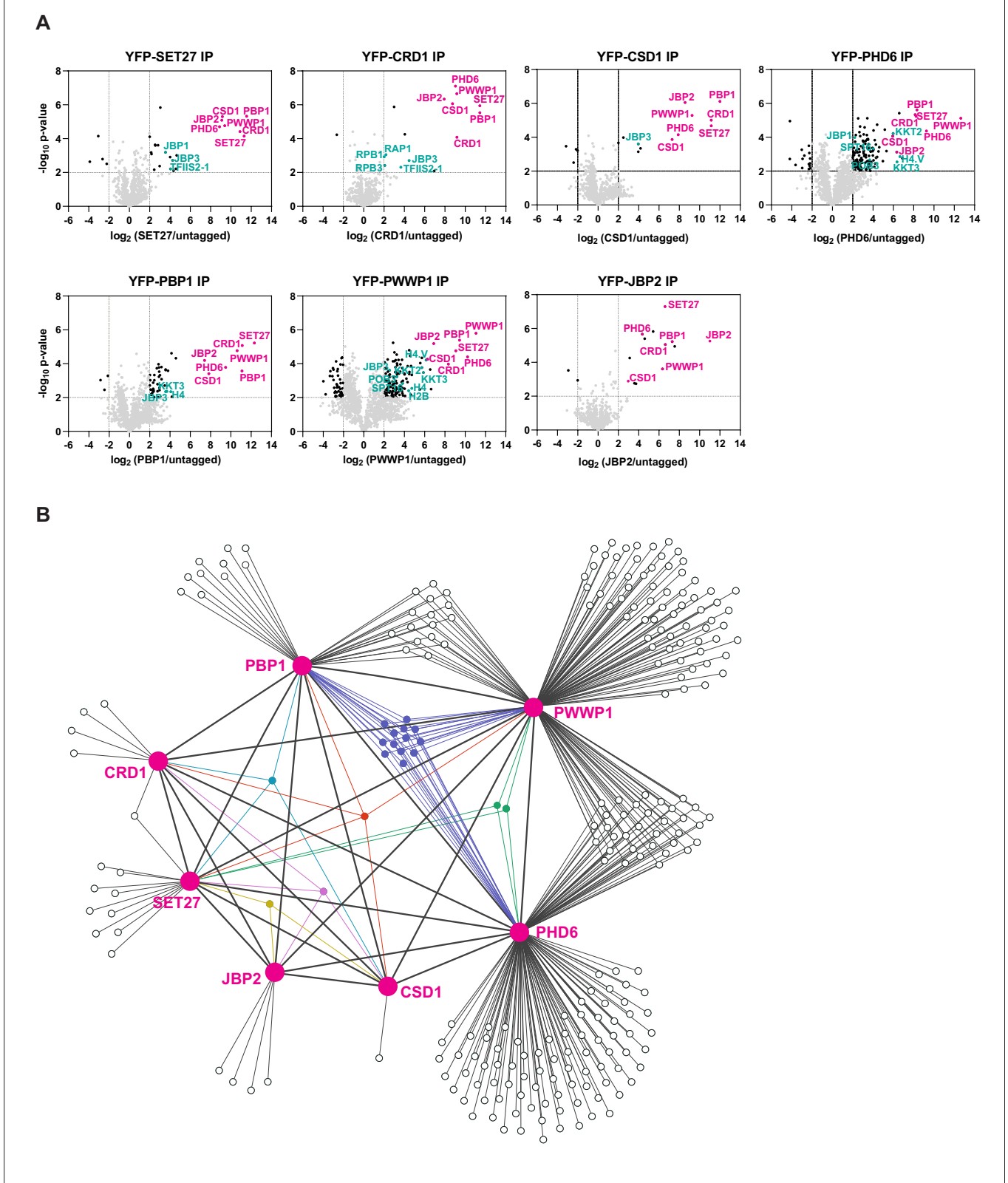

**Figure 1.** Identification of SPARC in bloodstream form *Trypanosoma brucei*. (**A**) Proteins previously shown to be enriched in CRD1 and SET27 coimmunoprecipitations (co-IPs) were YPF-tagged and analyzed by affinity selection and LC-MS/MS to identify their protein interaction partners. Volcano plots are based on three biological replicates for each sample. Cutoffs used for significance: $\log_2$ (tagged/untagged)>2 or <−2 and Student's t-test p-value<0.01. See ***Supplementary file 1*** for a complete list of proteins enriched in each affinity selection. Putative SPARC complex subunits are

*Figure 1 continued on next page*

*Figure 1 continued*

shown in pink and other proteins of interest are shown in teal. The CRD1 and SET27 co-IPs are reproduced from Figure 4A by *Staneva et al., 2021*. (**B**) Interaction network of the proteins enriched in the co-IP experiments shown in (**A**). SPARC components are connected with thick lines while all other interactions are shown with thin lines. Proteins which interact with three or more SPARC components are represented in different colors. See *Supplementary file 2* for a complete list of shared and unique interactors in these co-IPs.

The online version of this article includes the following figure supplement(s) for figure 1:

**Figure supplement 1.** Domain architecture and localization of SPARC complex subunits.

**Figure supplement 2.** SPARC is present in procyclic form cells.

cells, JBP2 was predominantly cytoplasmic, whereas SET27 was strongly enriched in the nucleus. This contrasts with BF cells, where SET27 and JBP2 were detected in both the nucleus and cytoplasm (*Figure 1—figure supplement 1B*). CRD1 localized to the nucleus in both developmental forms of *T. brucei*.

Affinity selections of YFP-tagged SET27 and CRD1 from PF cells showed that both proteins strongly associate with each other and with the other four core SPARC components. Indeed, CSD1, PHD6, PBP1, and PWWP1 were at least sixfold more enriched in YFP-SET27 and YFP-CRD1 purifications relative to any other protein (*Figure 1—figure supplement 2A*; *Supplementary file 1*). However, in contrast to BF cells, affinity-selected YFP-SET27 displayed only a weak association with JBP2 (50-fold less than other SPARC components) and YFP-CRD1 showed no interaction with JBP2

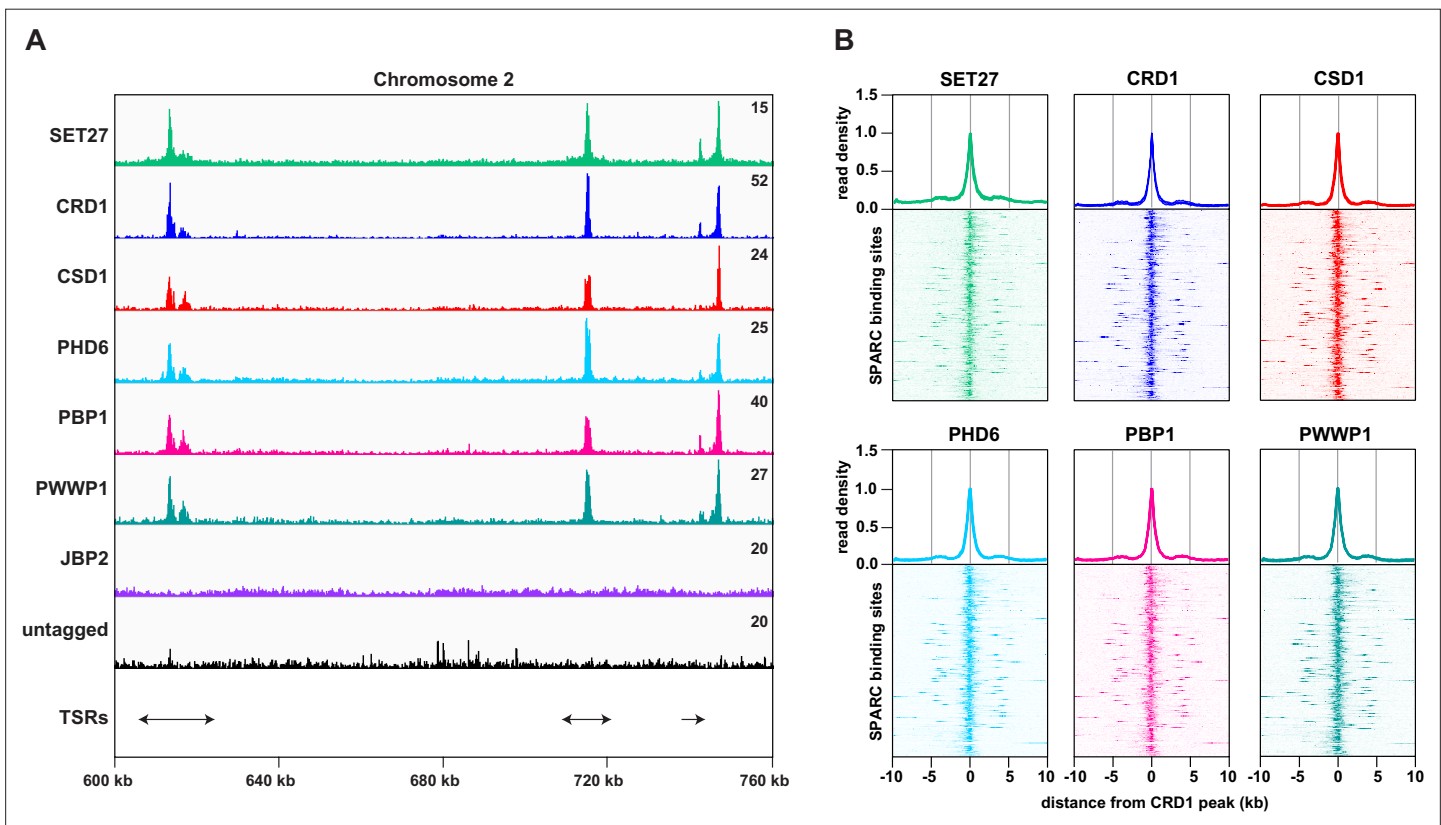

**Figure 2.** SPARC components target the same genomic loci in bloodstream form *Trypanosoma brucei*. (**A**) A region of Chromosome 2 is shown with ChIP-seq reads mapped for the indicated proteins. A single replicate is shown for each protein. ChIP-seq performed in wild-type cells lacking any tagged protein (untagged) was included as a negative control. Tracks are scaled separately as fragments per million (the exact value is indicated in the top-right corner of each track). The positions of single and double transcription start regions (sTSRs and dTSRs) are shown below with arrows indicating the direction of transcription. ChIP-seq data for CRD1, SET27, and the untagged parental cell line are reproduced from Figure 2A by *Staneva et al., 2021*. (**B**) Enrichment profiles for SPARC components. CRD1 is used as a reference because it has the most prominent peaks at TSRs. The metagene plots (*top*) show normalized read density around all CRD1 peak summits, with each replicate plotted separately. The heatmaps (*bottom*) show protein density around 177 individual CRD1 peaks. Each heatmap shows the average of at least two replicates. CRD1 and SET27 metagene plots and heatmaps are reproduced from Figure 2C by *Staneva et al., 2021*.

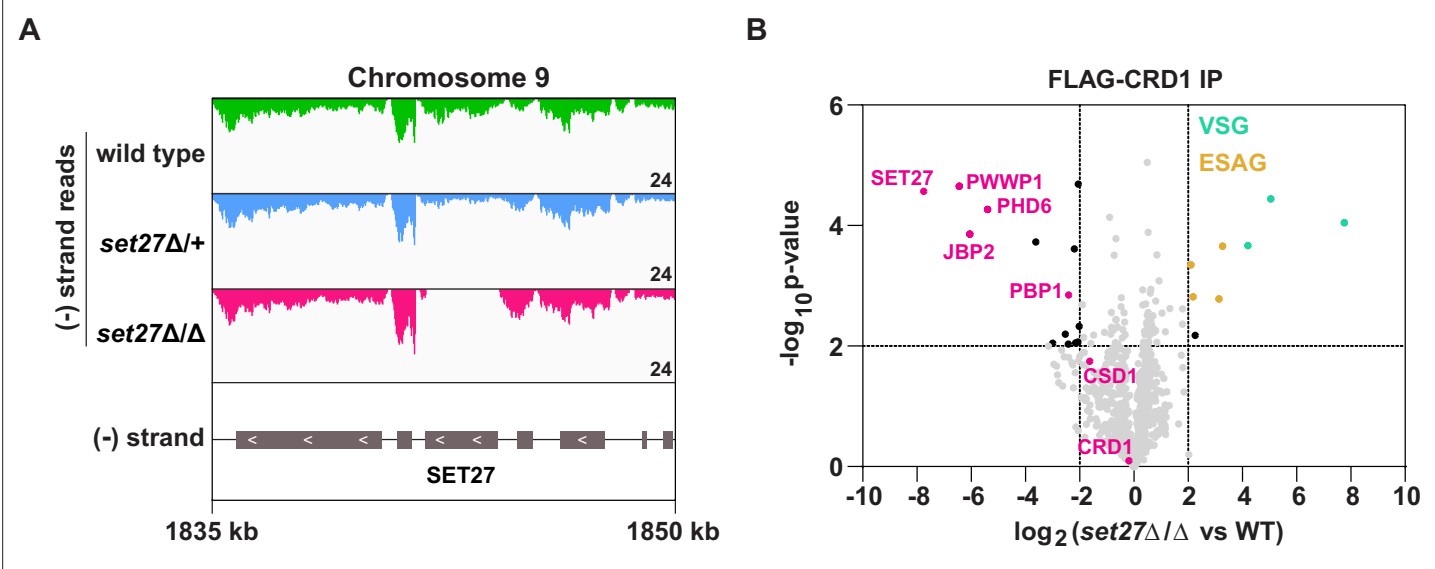

**Figure 3.** SPARC integrity is compromised in bloodstream form *T. brucei* lacking SET27. (**A**) Tracks showing the distribution of RNA-seq reads in the vicinity of the *SET27* gene in wild-type, *set27Δ/+* and *set27Δ/Δ* cells. All tracks are scaled identically (number of reads shown in the bottom right corner of each track). (**B**) Affinity selection of FLAG-CRD1 from wild-type and *set27Δ/Δ* cells. The volcano plot is based on three biological replicates for each sample. Cutoffs used for significance: $\log_2$ (*set27Δ/Δ* vs. WT)>2 or <−2 and Student's t-test p-value<0.01. See ***Supplementary file 1*** for a complete list of proteins in these affinity selections. SPARC components are shown in pink, VSGs in green, and ESAGs in yellow.

The online version of this article includes the following figure supplement(s) for figure 3:

**Figure supplement 1.** Validation of the SET27 knockout.

**Figure supplement 2.** Validation of the JBP2 knockout.

in PF trypanosomes (***Figure 1—figure supplement 2A***; ***Supplementary file 1***). Similarly, core SPARC components were only weakly enriched with affinity-selected YFP-JBP2 (~2-fold less compared to other proteins). This weak association of SPARC subunits with JBP2 in PF cells perhaps reflects its predominantly cytoplasmic localization relative to core components of the complex (***Figure 1—figure supplement 1B***).

We next performed ChIP-seq to map the genome-wide binding sites of YFP-tagged SET27, CRD1, and JBP2 in PF cells. As in BF cells, SET27 and CRD1 were both localized to RNAPII TSRs, whereas JBP2 was not chromatin-associated (***Figure 1—figure supplement 2B***).

These analyses in PF cells indicate that six core components, SET27, CRD1, CSD1, PHD6, PBP1, and PWWP1, form the main SPARC complex in both BF and PF cells. The ancillary factor JBP2 was largely absent from PF cell nuclei and only very weakly associated with other SPARC components in PF cells, underscoring its distinct behavior.

## Loss of SET27 disrupts SPARC formation

The strong association of SPARC with RNAPII TSRs in both BF and PF *T. brucei* cells suggested that it might have a role in regulating transcription. To test this, we attempted to delete the genes encoding SPARC components in BF cells. We successfully generated cell lines lacking the core subunit SET27 (*set27Δ/Δ*) and the auxiliary subunit JBP2 (*jbp2Δ/Δ*). However, we were unable to obtain cell lines completely null for any other SPARC component, suggesting these may be essential for viability. To confirm the *SET27* and *JBP2* gene deletions, we generated RNA-seq libraries using poly(A)-selected RNA. Cells lacking either one (Δ/+) or both alleles (Δ/Δ) showed a partial and complete loss of *SET27* and *JBP2* mRNA, respectively (***Figure 3A***; ***Figure 3—figure supplement 2A***). Importantly, transcripts derived from neighboring genes were unaffected, demonstrating that deletion of the *SET27* or *JBP2* genes did not alter the expression of other genes in their respective polycistrons.

Initial phenotypic characterization of cells lacking SET27 or JBP2 showed that *set27Δ/Δ* cells grew substantially slower than wild-type or *set27Δ/+* cells (***Figure 3—figure supplement 1A***). In contrast,

*jbp2Δ/Δ* cells grew slightly faster than wild-type cells (*Figure 3—figure supplement 2B*). The basis for these differences in growth rates remains to be determined.

To test if SET27 is required for the integrity of SPARC, we FLAG-tagged CRD1 at its endogenous locus in wild-type and *set27Δ/Δ* cells. Next, we performed affinity selections of FLAG-CRD1 from both cell lines followed by mass spectrometry to detect potential differences in interacting proteins. In the absence of SET27, FLAG-CRD1 showed greatly reduced association with all other SPARC components (*Figure 3B*; *Supplementary file 1*). Instead, several VSG proteins were detected as being associated with FLAG-CRD1 in *set27Δ/Δ* cells, though it is likely that these represent unspecific interactions. Thus, we conclude that SET27 is pivotal for SPARC assembly or stability, as other subunits dissociate in its absence.

## SET27 is required for correct transcription initiation in BF *T. brucei*

Because SPARC is enriched over transcription start regions, we tested whether SET27 is required for accurate transcription initiation. In *T. brucei*, transcription initiates from either unidirectional or bidirectional promoters. All regions annotated as sTSRs in the trypanosome genome correspond to unidirectional promoters which are associated with a single SPARC binding site. Regions annotated as dTSRs exhibit either one or two SPARC peaks depending on whether the promoter is bidirectional or there are two nearby unidirectional promoters from which transcription initiates in opposite directions (e.g., see *Figures 4 and 5*). In wild-type and *set27Δ/+* cells, transcription typically initiated coincident with, or just downstream from, the peak of SPARC binding (*Figure 4A–C*). Strikingly, this pattern was lost in *set27Δ/Δ* cells. At all unidirectional promoters, the transcription start site shifted ~1–5 kb upstream of its normal position in wild-type cells (*Figure 4A*, *left*), though the extent of this defect varied between promoters in terms of both distance from the wild-type start site and RNA amount. Additionally, at 14 out of 54 normally unidirectional promoters, transcription initiated in both directions in *set27Δ/Δ* cells, effectively converting them to bidirectional promoters (*Figure 4A*, *right*). These altered transcription patterns were also clearly evident in a metagene analysis of unidirectional promoters (*Figure 4C*). Data from dTSRs were more difficult to interpret because in these regions one cannot distinguish between an upstream and an antisense transcriptional defect. Nonetheless, there was strong evidence that SET27 also contributes to accurate transcription initiation from dTSRs (*Figure 4B, C*). To determine if the transcriptional phenotype we observed was specifically due to SET27 loss, we restored the wild-type *SET27* gene to its endogenous locus by homologous recombination (*Figure 3—figure supplement 1B*). We obtained two rescue clones which grew slower than wild-type but faster than *set27Δ/Δ* cells (*Figure 3—figure supplement 1A*). Importantly, the elevated level of transcription within two TSRs observed in *set27Δ/Δ* cells was completely reversed when the *SET27* gene was restored (*Figure 4D*).

In contrast to SET27, loss of JBP2 did not result in any distinct changes to the transcriptional profile around TSRs (*Figure 3—figure supplement 2C*). This is consistent with our ChIP-seq data showing that JBP2 does not selectively associate with promoter regions (*Figure 2*), and again indicates that its function is distinct from core SPARC components. Collectively, these results suggest that SPARC contributes to the accurate positioning of transcription start sites and to normal transcription directionality.

## The distribution of RNAPII is altered in cells lacking SET27

If promoter positioning is indeed affected by SET27 loss, then RNAPII occupancy within TSRs would also be expected to change. To directly test this possibility, we performed ChIP-seq for YFP-RPB1 (the largest RNAPII subunit) in wild-type, *set27Δ/+*, and *set27Δ/Δ* BF *T. brucei* cells. In wild-type and *set27Δ/+* cells, RPB1/RNAPII enrichment was clearly coincident with SPARC peaks (represented by CRD1; *Figure 5*). Notably, this pattern was significantly altered in *set27Δ/Δ* cells, with RPB1/RNAPII peaks becoming less defined and, in some instances, completely lost (*Figure 5A*). Comparison with RNA-seq data showed that the broader RPB1/RNAPII signal in *set27Δ/Δ* cells coincides with transcript accumulation upstream of the major transcription initiation site in wild-type cells (*Figure 5A*). Metagene analysis confirmed that the RPB1/RNAPII peaks are generally reduced and broader in *set27Δ/Δ* cells relative to *set27Δ/+* or wild-type cells (*Figure 5B*). Thus, we conclude that SET27 is required for accurate RNAPII transcription initiation.

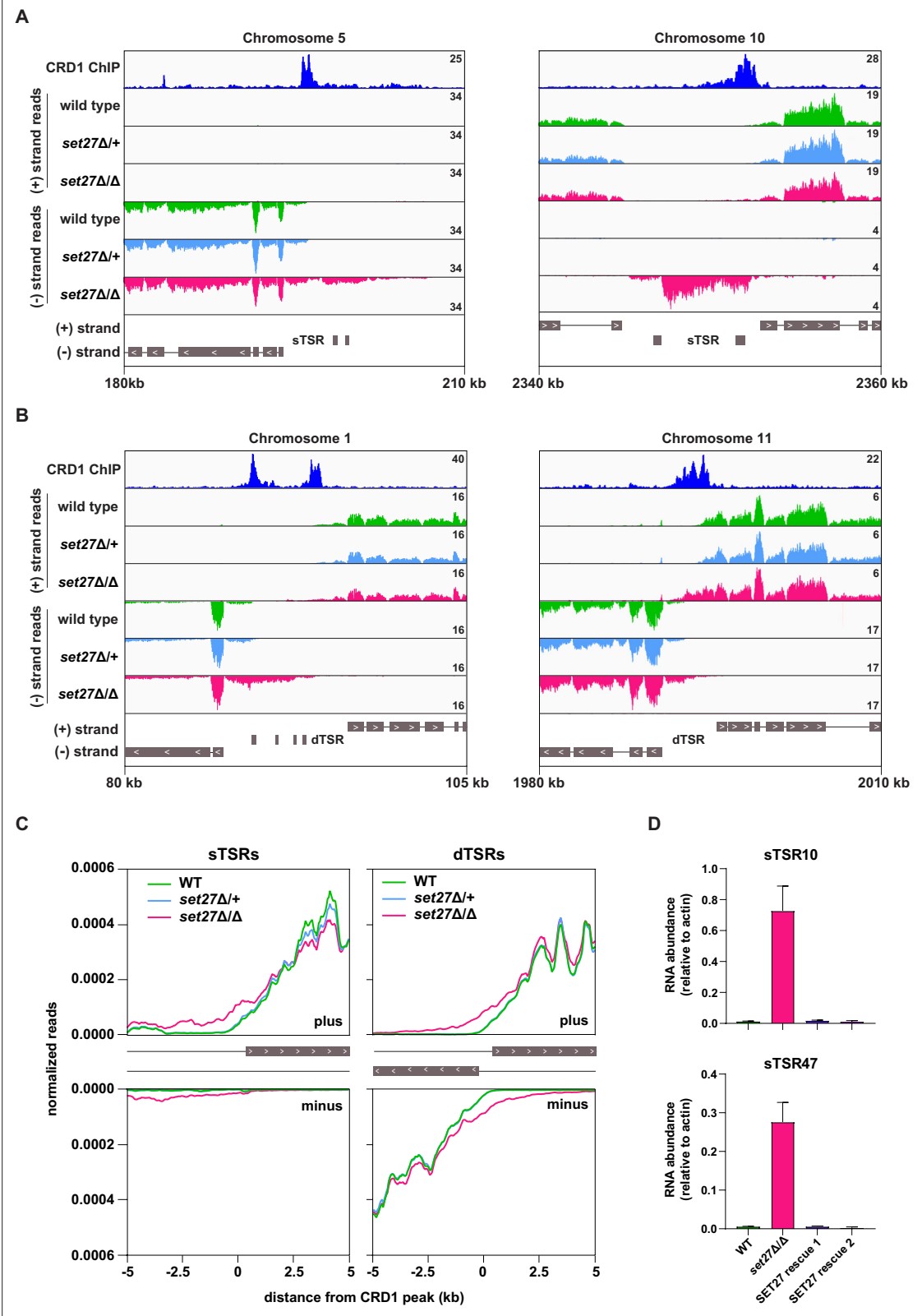

**Figure 4.** Transcription initiation is dysregulated in the absence of SET27. (**A**) Tracks showing the distribution of RNA-seq reads in the presence (wild-type and *set27Δ/+*) or absence of SET27 (*set27Δ/Δ*) around selected unidirectional sTSR promoters. CRD1 ChIP (*top track*) is included to mark the position of SPARC sites. ORFs are indicated by gray boxes, and directionality is shown with inset white arrows. Genes present within a single polycistron are connected with a thin black line. Hypothetical protein-coding genes annotated within each TSR region are not connected to neighboring

*Figure 4 continued on next page*

*Figure 4 continued*

polycistrons. (**B**) As in (**A**), but for dTSR promoters. (**C**) *Left*: metaplots showing the distribution of RNA-seq reads around 33 SPARC sites marking unidirectional sTSR promoters. For clarity, we excluded SPARC sites present within 5 kb of a different SPARC site. Transcription in the forward direction is shown in the upper panel and transcription in the reverse direction is shown below. *Right*: metaplots showing the distribution of RNA-seq reads surrounding all 120 SPARC sites present at dTSR promoters. (**D**) RT-qPCR of RNA originating from sTSR10 and sTSR47 in wild-type, *set27Δ/Δ*, and two SET27 rescue clones. Error bars: standard deviation (SD) of two biological replicates.

## SET27 represses the expression of VSG and transposon-derived genes

The finding that transcription initiation and RNAPII occupancy is altered in *set27Δ/Δ* cells prompted us to determine if these changes affect the expression of any mRNAs. Principal component analysis was employed to compare the global transcriptional landscapes of wild-type, *set27Δ/+*, and *set27Δ/Δ* cells (*Figure 6A*). Wild-type and *set27Δ/+* replicates clustered together while *set27Δ/Δ* replicates were clearly distinct, indicating that gene expression is aberrant in cells lacking SET27. Analysis of differentially expressed genes (DEGs) in wild-type versus *set27Δ/Δ* cells revealed widespread transcriptional induction of otherwise silent or weakly expressed genes (*Figure 6B, C*). Almost 100 mRNAs were significantly upregulated at least twofold (*Figure 6B*; *Supplementary file 3*). Conversely, only the *SET27* mRNA was reduced, due to its gene deletion (*Figure 6B*; *Supplementary file 3*). Gene ontology analysis of the upregulated mRNA set revealed strong enrichment for normally silent VSG genes (*Figure 6B–D*) which were distinct from the VSG proteins detected in FLAG-CRD1 immunoprecipitations from *set27Δ/Δ* cells (*Figure 3B*). Additionally, we observed derepression of SLACS retrotransposons which reside within the SL gene cluster, as well as derepression of reverse transcriptase and RNase H genes derived from the more widely distributed ingi retrotransposons (*Figure 6B–D*). Expression site-associated genes (ESAGs; transcribed together with the active VSG) were also upregulated in *set27Δ/Δ* cells (*Figure 6B, C*). However, the ESAG gene category was not significantly enriched overall among the DEGs (*Figure 6D*).

To determine if the DEGs identified as being upregulated in *set27Δ/Δ* cells share any particular feature that might explain their sensitivity to SPARC loss, we mapped their genomic location relative to the nearest SPARC peak. Genes whose exact genomic location varies, such as VSG genes that are located at subtelomeric regions and can frequently recombine to new locations, were excluded from this analysis. Notably, upregulated genes typically resided within 2.5 kb of a SPARC peak, compared to 65 kb for the average, unaltered gene (*Figure 6E*).

A sizable subset of DEGs were VSG genes located at the ends of chromosomes adjacent to telomeres. Interestingly, upon manual inspection, we noticed that a large number of these induced subtelomeric VSG transcripts (16 out of 37) were derived from two locations on chromosome 9 near the right telomere. In both cases, these VSG gene transcripts arise downstream of SPARC peaks but are not immediately adjacent to them (*Figure 6F*). In the absence of SET27, RPB1/RNAPII displayed increased chromatin association over a much larger region downstream of each SPARC peak: ~60 and ~130 kb regions containing 11 and 5 differentially expressed VSGs, respectively (*Figure 6F*). Two non-VSG genes were also upregulated in this region, indicating that general derepression occurs across this subtelomere. We also observed a similar phenotype at other subtelomeric regions, such as Chromosome 8_5A where 10 VSGs and a gene encoding a hypothetical protein were upregulated upon SET27 deletion (*Supplementary file 3*).

These analyses demonstrate that SPARC is required to restrict the expression level of genes in the immediate vicinity of TSRs located within the main body of *T. brucei* chromosomes as well as genes, particularly VSGs, located further downstream of SPARC sites within subtelomeric regions.

## Discussion

In this study, we identified SPARC, a promoter-associated protein complex in both BF and PF of *T. brucei*, comprising SET27, CRD1, CSD1, PHD6, PBP1, and PWWP1. Several SPARC components showed homology to known histone mark reader and writer domains, including SET, Chromo, Chromoshadow, PHD, and PWWP, indicating that the association of SPARC with chromatin is mediated through binding to specific histone modification(s). JBP2 also copurified with core SPARC components but it appears to be functionally distinct from them with respect to chromatin binding and impact on transcription initiation. In contrast to a previous report (*DiPaolo et al., 2005*), we detected YFP-JBP2

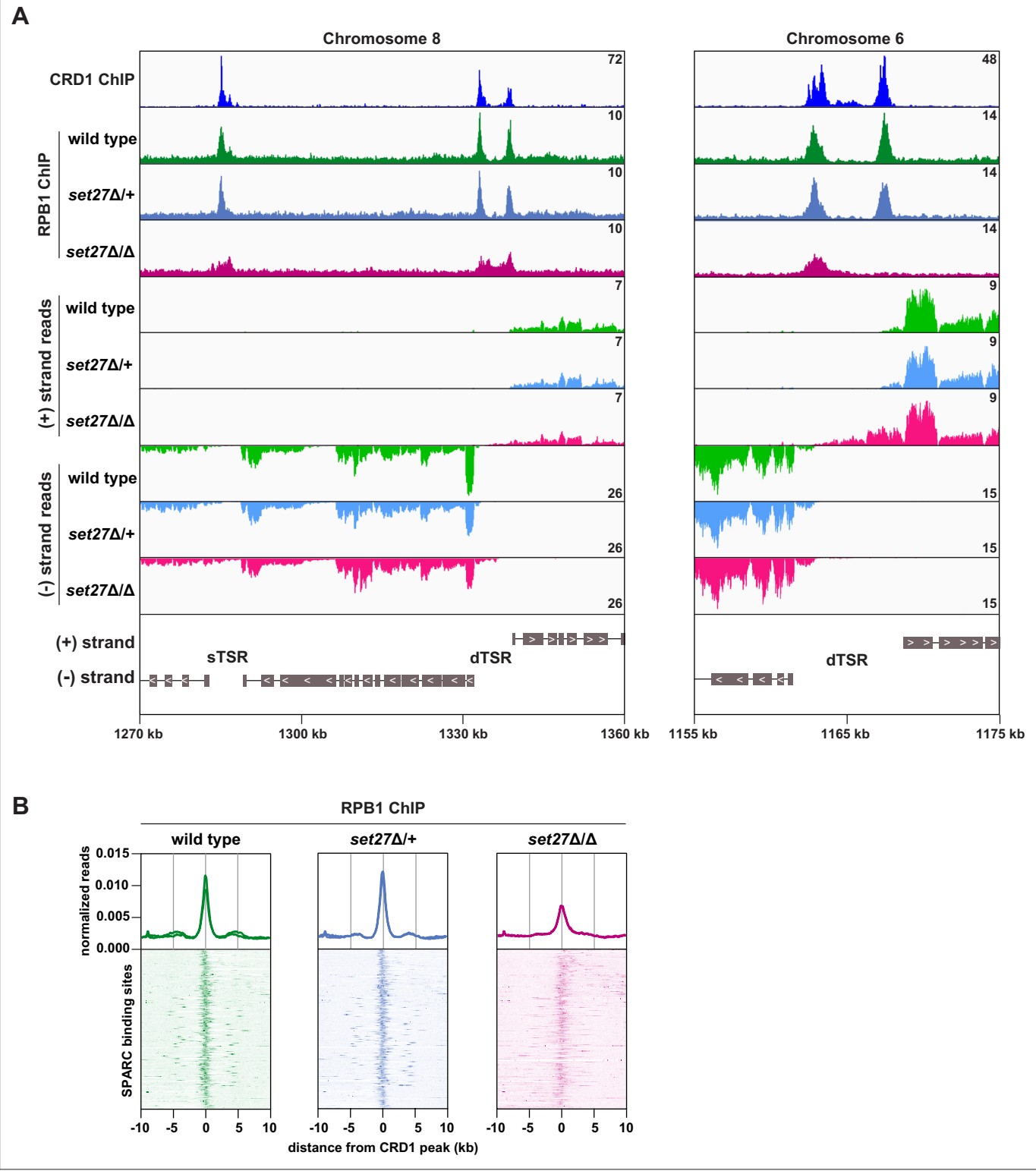

**Figure 5.** SET27 is required for full RNAPII recruitment to transcription start regions. (**A**) Tracks showing the distribution of YFP-tagged RPB1, the largest subunit of *T. brucei* RNAPII, across selected genomic windows following ChIP-seq. CRD1 ChIP (*top*) and RNA-seq (*bottom*) tracks are included for comparison. (**B**) RPB1 enrichment profiles. CRD1 is used as a reference because it has the most prominent peaks at TSRs. The metagene plots (*top*) show normalized RPB1 read density around all SPARC sites, with each replicate plotted separately. The heatmaps (*bottom*) show RPB1 density around 177 individual SPARC sites. Each heatmap shows the average of at least two replicates.

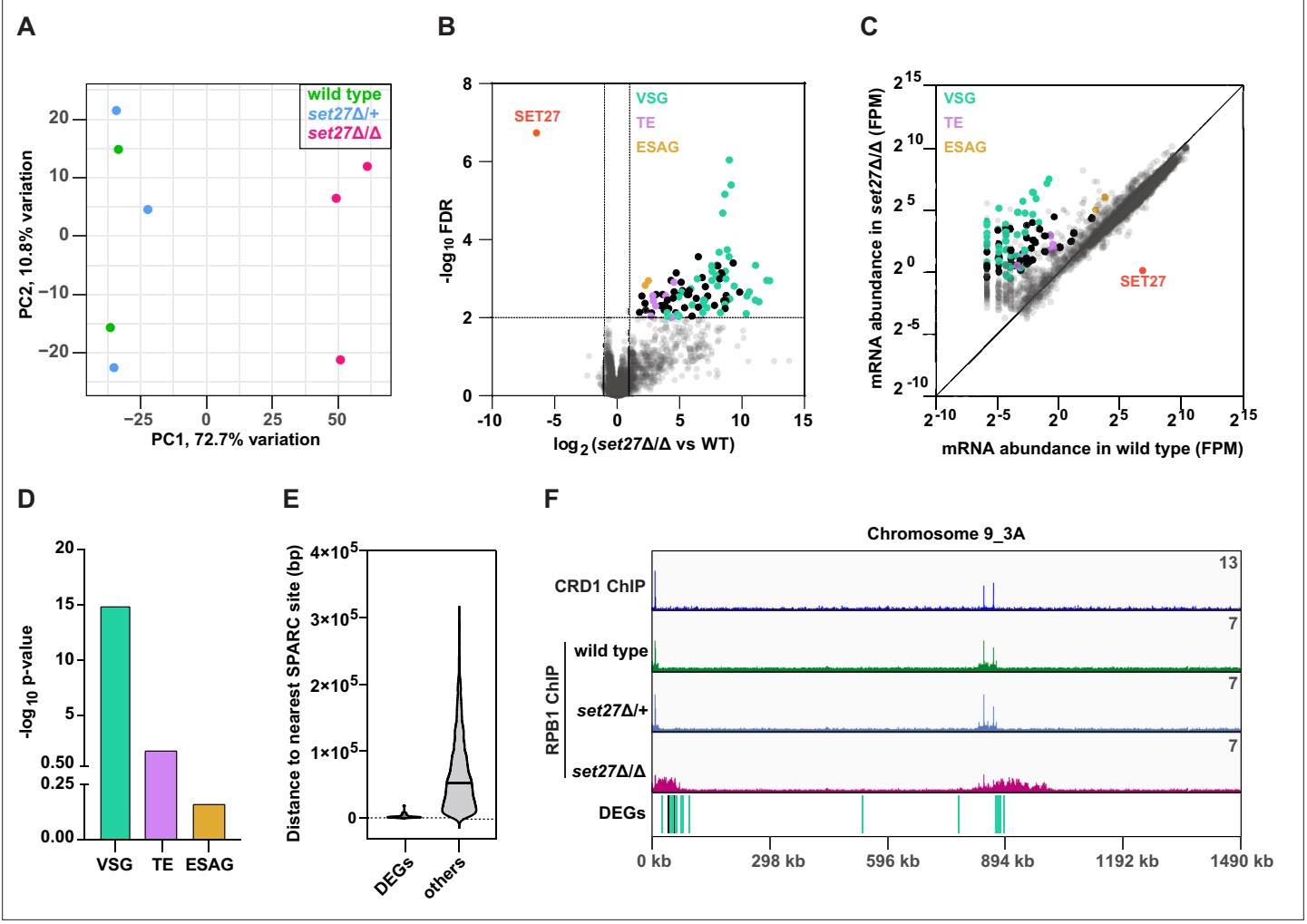

**Figure 6.** SET27 deletion derepresses VSGs and transposon-associated genes. (**A**) Principal component analysis (PCA) comparing mRNA expression (n=8540) in wild-type, *set27Δ/+*, and *set27Δ/Δ* cells. (**B**) Volcano plot showing differentially expressed genes (DEGs) between wild-type and *set27Δ/Δ* cells. Cut-offs used for significance: $\log_2$ (tagged/untagged)>1 or <−1 and FDR<0.01. VSGs are colored in green; SLACS and ingi transposable element (TE) genes are in lavender; ESAG genes are in yellow; other DEGs are in black; and non-DEGs are in gray. See **Supplementary file 3** for a complete list of DEGs. (**C**) Scatter plot of mRNA abundance normalized as FPM (fragments per million) in wild-type versus *set27Δ/Δ* cells. The diagonal marks the position of genes with equal expression in the wild-type and *set27Δ/Δ* cell lines. The color scheme is the same as in (**B**). (**D**) Gene ontology enrichment among upregulated mRNAs. P-values were calculated using Fisher's exact test. (**E**) Violin plot showing the distance between the 5′ end of each DEG and the nearest SPARC site. Non-DEGs (others) are included as a comparison. Only genes found in the main body of chromosomes were included in this analysis. Subtelomeric genes were excluded because they frequently recombine and their precise location is uncertain. (**F**) The Chromosome 9_3A subtelomeric contig showing ChIP-seq reads for CRD1 and RPB1. The locations of DEGs are shown below. The color scheme is the same as in (**B**).

and identified untagged JBP2 as a YFP-SET27 interactor in PF cells, indicating that JBP2 is expressed in PF *T. brucei*.

A chromatin-associated complex containing homologs of SET27, CRD1, CSD1, and PBP1 was recently identified in *Leishmania* (**Jensen et al., 2021**), showing that SPARC-related complexes are also present in other kinetoplastids, albeit with potentially altered composition. The *Leishmania* complex subunits interacted strongly with JBP3, a DNA base J binding protein, and deletions of JBP3 were shown to result in transcriptional read-through downstream of RNAPII termination sites in both *T. brucei* and *Leishmania* (**Jensen et al., 2021**; **Kieft et al., 2020**). Similarly, we detected JBP3 in affinity purifications of most core SPARC components. However, neither SET27 nor JBP2 deletions resulted in transcription termination defects, indicating that JBP3 function is distinct from SPARC.

While most SPARC subunits appear to be essential for viability, we were able to delete both alleles of the *SET27* gene, leading to dissociation of the other complex subunits. In the absence

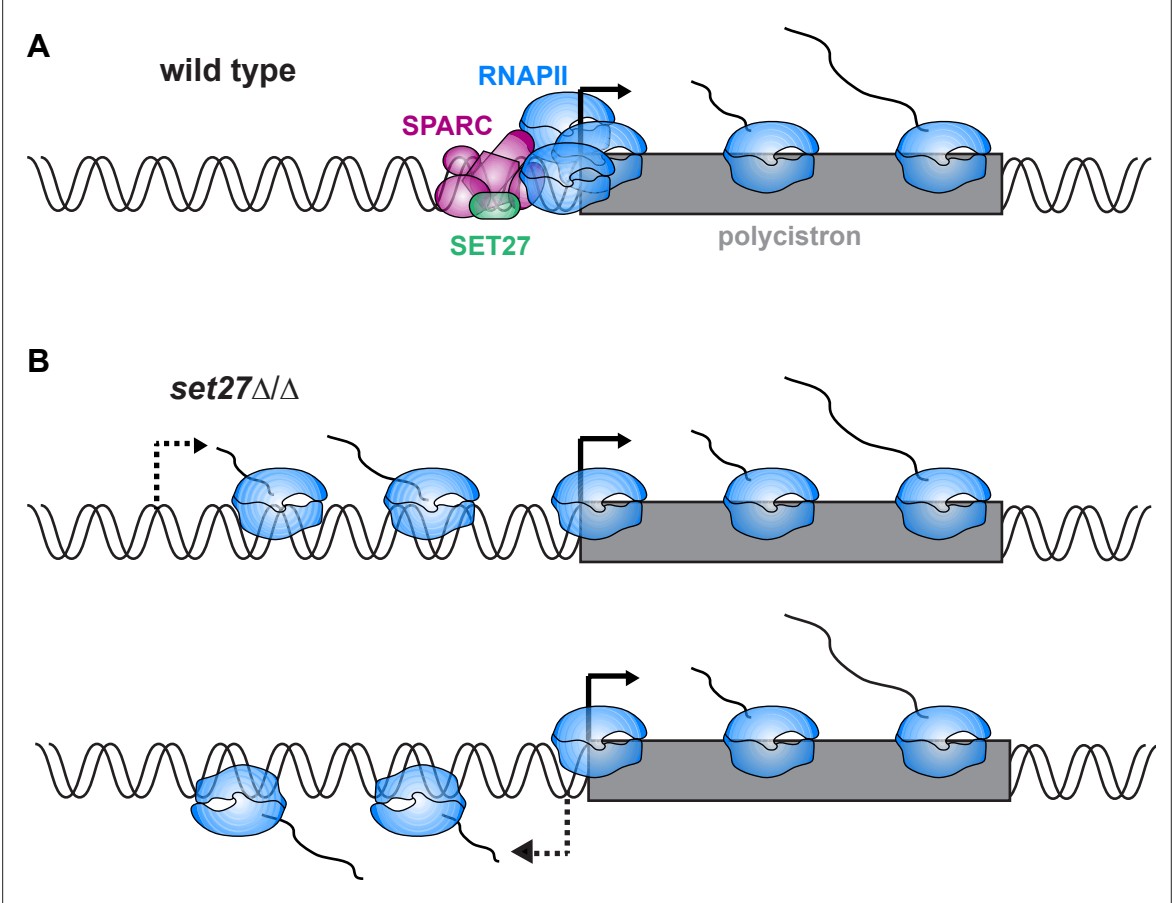

**Figure 7.** Model for SPARC-mediated definition of RNAPII transcription start sites. (**A**) In wild-type cells, SPARC associates with genomic sites just upstream of polycistronic transcription units. SPARC sites coincide with promoters and regions of RNAPII enrichment. (**B**) In the absence of SET27, the SPARC complex dissociates from promoters and RNAPII enrichment is decreased. Transcription initiates upstream of the natural site (*top*) and sometimes in the reverse direction (*bottom*), effectively converting some unidirectional promoters into bidirectional promoters.

of SET27, polyadenylated transcripts in the vicinity of SPARC binding sites were upregulated as a result of transcript accumulation upstream of the wild-type start site and antisense transcription from some normally unidirectional promoters. In principle, these observations are consistent with SPARC regulating either RNA synthesis or degradation. However, the balance of evidence suggests SPARC directly regulates transcription. First, our ChIP-seq data sets showed that SPARC subunits associate with RNAPII promoters in a narrow region of enrichment that matches that of RPB1. These similar enrichment patterns suggest a possible direct link between SPARC and the RNAPII transcription machinery of *T. brucei*. Such a link is supported by the interactions of CRD1, PHD6, and PWWP1 with RNAPII subunits in our proteomics analyses (*Figure 1*). In contrast, we did not observe any interactions between SPARC subunits and RNA decay factors. Second, SET27 loss disrupted RPB1 association with TSRs, suggesting SPARC directly or indirectly regulates transcription initiation. Taken together, these results support the conclusion that SPARC functions at *T. brucei* promoters by interacting with RNAPII to accurately demarcate transcription start sites and, in some instances, ensure unidirectional transcription (*Figure 7*). Sequence-specific elements have recently been found to drive RNAPII transcription from a *T. brucei* promoter (*Cordon-Obras et al., 2022*); however, we were unable to identify similar motifs underlying CRD1 or RPB1 ChIP-seq peaks, suggesting that *T. brucei* promoters are perhaps heterogeneous in composition.

While SPARC contributes to the accuracy of transcription initiation across essentially all RNAPII promoters, the abundance of most mRNAs was unaffected following SET27 loss. This probably reflects the dominant role of post-transcriptional regulation in *T. brucei* gene expression (*Clayton, 2019*). Such post-transcriptional buffering might have evolved primarily to regulate expressed mRNAs, leaving

regions close to promoters, which are not normally transcribed, free from this regulation. This may explain why increased transcript levels in *set27Δ/Δ* were restricted to the vicinity of TSRs. However, widespread transcriptional dysregulation of mRNAs and its post-transcriptional correction might be energetically costly, potentially leading to the growth defect observed in *SET27* null cells.

Of the genes that changed expression after SET27 deletion, almost half belong to the VSG family of surface coat proteins. BF trypanosomes normally express only a single VSG gene from 1 of ~15 telomere-adjacent bloodstream expression sites (BESs). In contrast, in *set27Δ/Δ* cells, we detected upregulation of 43 VSG transcripts, none of which were annotated as located in BES regions. Recently, *López-Escobar et al., 2022* have also observed VSG mRNA upregulation from non-BES locations, suggesting that VSGs might sometimes be transcribed from other regions of the genome. However, the VSG transcripts we detect as upregulated in *set27Δ/Δ* were relatively lowly expressed (*Figure 6C*) and may not be translated to protein or be translated at low levels compared to a VSG transcribed from a BES site. Most of the upregulated VSG transcripts originated from subtelomeric regions, far from SPARC binding sites. We propose that, in contrast to promoters, transcriptional silencing within subtelomeric regions is sensitive to loss of SPARC activity over large chromosomal regions. However, an alternative possibility is that transcriptional repression in subtelomeric regions is mediated by different protein complexes which share some of their subunits with SPARC, or whose activity is influenced by it.

Additionally, in cells lacking SET27, we observed derepression of promoter-proximal transcripts derived from retrotransposons. It is possible that a more open chromatin configuration at trypanosome TSRs allows preferential insertion of transposable elements at those locations, similar to what has been observed in other eukaryotes (*Feschotte, 2008*; *Miao et al., 2020*). In wild-type cells, SPARC may help maintain genome integrity by preventing the expression of these mobile genetic elements.

In summary, we have identified SPARC, a novel promoter-associated protein complex that helps define RNAPII transcription start sites, influences transcription directionality, and represses transcription across subtelomeric domains. The function of the SET27 putative histone methyltransferase at *T. brucei* TSRs is reminiscent of the role that the Set1/COMPASS complex plays at Opisthokont RNAPII promoters where it writes the H3K4me mark (*Shilatifard, 2012*). H3K4 methylation by Set1/COMPASS is mediated through a PHD reader module and interactions with RNAPII, influencing cryptic transcript degradation and both activation and repression of RNAPII transcription (*Howe et al., 2017*). Despite the divergence of histones in trypanosomes, they also exhibit H3K4 methylation that is enriched over promoter-proximal regions (*Kraus et al., 2020*; *Wright et al., 2010*). It remains to be determined which histone lysine residues, if any, are methylated by SET27, which reader proteins (i.e., CRD1, PHD6, and/or PWWP1) bind the resulting modification(s) and if their binding also contributes to promoter definition.

# Materials and methods

**Key resources table**

| Reagent type (species) or resource | Designation | Source or reference | Identifiers | Additional information |
|---|---|---|---|---|
| Gene (*Trypanosoma brucei*) | SET27 | TriTrypDB | Tb927.9.13470 | |
| Gene (*T. brucei*) | CRD1 | TriTrypDB | Tb927.7.4540 | |
| Gene (*T. brucei*) | CSD1 | This paper | Tb927.11.11840 | |
| Gene (*T. brucei*) | PHD6 | This paper | Tb927.3.2350 | |
| Gene (*T. brucei*) | PBP1 | This paper | Tb927.1.4250 | |
| Gene (*T. brucei*) | PWWP1 | This paper | Tb927.11.13820 | |
| Gene (*T. brucei*) | JBP2 | TriTrypDB | Tb927.7.4650 | |
| Gene (*T. brucei*) | RPB1 | TriTrypDB | Tb927.4.5020 | |
| Strain, strain background (*T. brucei*) | Lister 427 | Keith Matthews lab stocks | | BF and PF |
| Cell line (*T. brucei*) | YFP-SET27 | *Staneva et al., 2021* | | BF |

*Continued on next page*

*Continued*

| Reagent type (species) or resource | Designation | Source or reference | Identifiers | Additional information |
|---|---|---|---|---|
| Cell line (*T. brucei*) | YFP-CRD1 | *Staneva et al., 2021* | | BF |
| Cell line (*T. brucei*) | YFP-CSD1 | This paper | | BF |
| Cell line (*T. brucei*) | YFP-PHD6 | This paper | | BF |
| Cell line (*T. brucei*) | YFP-PBP1 | This paper | | BF |
| Cell line (*T. brucei*) | YFP-PWWP1 | This paper | | BF |
| Cell line (*T. brucei*) | YFP-JBP2 | This paper | | BF |
| Cell line (*T. brucei*) | YFP-SET27 | This paper | | PF |
| Cell line (*T. brucei*) | YFP-CRD1 | This paper | | PF |
| Cell line (*T. brucei*) | YFP-JBP2 | This paper | | PF |
| Cell line (*T. brucei*) | YFP-RPB1 | *Staneva et al., 2021* | | BF |
| Cell line (*T. brucei*) | *set27Δ/+* in YFP-RPB1 | This paper | | BF |
| Cell line (*T. brucei*) | *set27Δ/Δ* in YFP-RPB1 | This paper | | BF |
| Cell line (*T. brucei*) | *jbp2Δ/+* in J1339 WT | This paper | | BF |
| Cell line (*T. brucei*) | *jbp2Δ/Δ* in J1339 WT | This paper | | BF |
| Cell line (*T. brucei*) | FLAG-CRD1 in WT | This paper | | BF |
| Cell line (*T. brucei*) | FLAG-CRD1 in *set27Δ/Δ* | This paper | | BF |
| Transfected construct (*T. brucei*) | BSR-YFP contruct | This paper | | Used to generate the YFP-tagged cell lines |
| Transfected construct (*T. brucei*) | PUR-FLAG construct | This paper | | Used to generate the FLAG-tagged cell lines |
| Transfected construct (*T. brucei*) | SET27/JBP2 5′ UTR - HYG/G418 - SET27/JBP2 3′ UTR | This paper | | Used to generate the *set27* and *jbp2* Δ/+ and Δ/Δ cell lines |
| Transfected construct (*T. brucei*) | SET27 5′ UTR - PUR - SET27 CDS - SET27 3′ UTR | This paper | | Used to generate the SET27 rescue cell line |
| Antibody | Anti-GFP (Rabbit polyclonal) | Thermo Fisher Scientific | Cat # A-11122 | IF (1:500) ChIP (10 µg per sample) |
| Antibody | Anti-GFP (Mouse monoclonal) | Roche | Cat # 11814460001 | IP-MS (10 µg per sample) |
| Antibody | M2 anti-FLAG (Mouse monosclonal) | Sigma-Aldrich | Cat #F1804 | IP-MS (10 µg per sample) |
| Antibody | Alexa Fluor 568 anti-rabbit (Goat polyclonal) | Thermo Fisher Scientific | Cat #A-11036 | IF (1:1000) |
| Recombinant DNA reagent | pPOTv4 BSR YFP (plasmid) | *Dean et al., 2015* | | Used for the YFP tagging |
| Recombinant DNA reagent | pPOTv4 PUR FLAG (plasmid) | This paper | | Used for the FLAG tagging |
| Recombinant DNA reagent | pPOTv7 G418 mNG (plasmid) | Gift from Sam Dean | | Used to generate the *set27* and *jbp2* Δ/+ and Δ/Δ cell lines |

*Continued*

| Reagent type (species) or resource | Designation | Source or reference | Identifiers | Additional information |
|---|---|---|---|---|
| Recombinant DNA reagent | pPOTv7 HYG RFP (plasmid) | Gift from Sam Dean | | Used to generate the *set27* and *jbp2* Δ/+ and Δ/Δ cell lines |
| Recombinant DNA reagent | pJ1339 (plasmid) | *Alves et al., 2020* | | Used to generate the *jbp2* Δ/+ and Δ/Δ cell lines |
| Recombinant DNA reagent | pMA-RQ-SET27 addback (plasmid) | Invitrogen | | Used to generate the SET27 rescue cell line |
| Commercial assay or kit | Quick Blunting Kit | NEB | Cat #E1201L | |
| Commercial assay or kit | Klenow Fragment (3′→5′ exo-) | NEB | Cat #M0212S | |
| Commercial assay or kit | NEXTflex-96 DNA Barcodes (Illumina Compatible) | Bioo Scientific | Cat #514105 | |
| Commercial assay or kit | Luna Universal One-Step RT-qPCR Kit | NEB | Cat #E3005S | |
| Commercial assay or kit | NEBNext Ultra II Directional RNA Library Prep Kit for Illumina | NEB | Cat #7760 | |
| Commercial assay or kit | NEBNext Poly(A) mRNA Magnetic Isolation Module | NEB | Cat #E7490 | |
| Chemical compound, drug | AMPure XP beads | Beckman Coulter | Cat #A63881 | |
| Chemical compound, drug | TRIzol reagent | Thermo Fisher Scientific | Cat # 15596026 | |
| Chemical compound, drug | RapiGest SF Surfactant | Waters | Cat #186001861 | |
| Chemical compound, drug | Trypsin Protease, MS Grade | Thermo Fisher Scientific | Cat #90057 | |
| Software, algorithm | Cytoscape | *Shannon et al., 2003* | | |
| Software, algorithm | LeishGEdit | *Beneke et al., 2017* | | |
| Software, algorithm | Perseus | *Tyanova et al., 2016* | | |
| Software, algorithm | TriTrypDB | *Aslett et al., 2010* | | |

## Cell lines

This study utilized two standard parental cell lines of the eukaryotic parasite *T. brucei*: BF 427 and PF 427. Analysis of ChIP-seq input and RNA-seq data made from these cell lines showed no evidence of contamination with sequence reads from *Mycoplasma*. Our sequencing data also confirmed that all cell lines are *T. brucei* 427. The cell lines with tagged proteins were confirmed by immunoprecipitation followed by mass spectrometry. The gene knockout cell lines were confirmed by RNA-seq. The SET27 rescue cell line was confirmed by RT-qPCR.

## Cell culture

BF *T. brucei* 427 parasites were grown at 37°C and 5% $CO_2$ in HMI-9 medium supplemented with Penicillin-Streptomycin (Gibco) and 10% fetal bovine serum (Gibco). PF *T. brucei* 427 cells were grown at 27°C in SDM-79 medium supplemented with hemin, Penicillin-Streptomycin (Gibco), and 10% FBS (Gibco).

## Structural bioinformatics

The presence of putative structural/functional domains in the SPARC complex subunits was identified by analyzing: (1) the amino acid sequence composition of individual subunits for remote protein homology using the HHPred web server (*Söding et al., 2005*), offered by the MPI Bioinformatics Tool kit (https://toolkit.tuebingen.mpg.de/tools/hhpred; *Zimmermann et al., 2018*; *Gabler et al.,*

*2020*), (2) the three-dimensional models generated by AlphaFold (*Jumper et al., 2021*), an artificial intelligence (AI) based high accuracy structure prediction method, using ColabFod notebook on Google CoLab (*Mirdita et al., 2022*). AlphaFold generated three-dimensional models were further analyzed using the DALI server (http://ekhidna2.biocenter.helsinki.fi/dali/; *Holm and Rosenström, 2010*) a webserver that compares a query protein structure against the available protein structures in the Protein Data Bank (PDB) to strengthen the conclusions of the HHpred analyses.

## Protein tagging

Proteins were tagged N-terminally at their endogenous loci using the pPOTv4 BSR YFP plasmid (*Dean et al., 2015*) and a derivative of the plasmid made by changing the blasticidin (BSR) drug resistance marker to puromycin (PUR) and the YFP tag to FLAG. Tagging constructs were made by fusion PCR and consisted of the end of the 5′ UTR of each gene, a region of the pPOTv4 plasmid containing the drug resistance gene and the tag, and the beginning of the CDS of each gene. Primers used to make the tagging constructs are listed in *Supplementary file 4*. Fusion constructs were transfected into *T. brucei* by electroporation. Cell lines with tagged proteins were verified by anti-GFP or anti-FLAG immunoprecipitation followed by LC-MS/MS.

## Generating knockout cell lines

The *SET27* gene deletion was performed in a YFP-RPB1 cell line (WT) via homologous recombination to replace both *SET27* alleles with hygromycin (HYG) and G418 drug resistance markers. The *JBP2* gene deletion was made using a CRISPR/Cas9 toolkit developed for kinetoplastids (*Beneke et al., 2017*). Briefly, a pJ1339 plasmid from which Cas9 is constitutively expressed (*Alves et al., 2020*) was integrated into the tubulin locus to generate the J1339 WT cell line. Then, guide RNAs and repair cassettes were transfected into the J1339 WT cell line to replace both *JBP2* alleles with HYG and G418 drug resistance markers. Oligos used to make these deletions are listed in *Supplementary file 4*. HYG and G418 resistance markers were amplified from pPOTv7 plasmids kindly gifted by Sam Dean. The knockout cell lines were verified by RNA-seq.

## Generating the SET27 rescue cell line

A pMA-RQ plasmid containing the rescue construct consisting of a fragment from the SET27 5′ UTR, followed by a puromycin (PUR) resistance cassette, the SET27 CDS sequence and a fragment from the SET27 3′ UTR were synthesized by Invitrogen. The plasmid was linearized by PsiI (NEB) and transfected by electroporation into *set27Δ/Δ* cells. The rescue clones obtained were verified by RT-qPCR.

## Immunolocalization

Parasites were harvested by centrifugation, washed once with phosphate-buffered saline (PBS) and fixed with 4% paraformaldehyde (final concentration) for 10 min on ice. Fixation was then stopped by adding glycine. Fixed cells were pipetted onto polylysine slides and permeabilized with 0.1% Triton X-100. Slides were blocked with 2% bovine serum albumin (BSA) for 45 min at 37°C. Blocked slides were incubated for 45 min at 37°C first with rabbit anti-GFP primary antibody (A-11122; Thermo Fisher Scientific) diluted 1:500 in 2% BSA, and then with Alexa fluor 568 goat anti-rabbit secondary antibody (A-11036; Thermo Fisher Scientific) diluted 1:1000 in 2% BSA. DNA was stained with DAPI. Slides were imaged using a Zeiss Axio Imager microscope.

## ChIP-seq

$3–5×10^8$ cells were harvested by centrifugation and fixed with 0.8% formaldehyde (final concentration) for 20 min at room temperature. Cells were lysed and chromatin was sheared by sonication in a Bioruptor sonicator (Diagenode) for 30 cycles (each cycle: 30 s ON, 30 s OFF; high setting). Sonication was performed in the presence of 0.2% SDS, after which sonicated samples were centrifuged to pellet cell debris and SDS in the supernatants was diluted to 0.07%. An input (0.8% of total sample) was taken before incubating the rest of the supernatant overnight with Protein G Dynabeads and 10 μg anti-GFP antibody (A-11122; Thermo Fisher Scientific). The following day, the beads were washed first in the presence of 500 mM NaCl, and then in the presence of 250 mM LiCl. The DNA was eluted from the beads using a buffer containing 50 mM Tris-HCl pH 8, 10 mM EDTA, and 1% SDS. The eluted sample was treated with DNase-free RNase (Roche) and Proteinase K, and DNA was purified

using a QIAquick PCR Purification Kit (Qiagen). ChIP-seq libraries were prepared as follows. DNA was blunt-ended using a Quick blunting kit (NEB), followed by treatment with Klenow Fragment 3'→5' exo- (NEB) and dATP. NEXTflex barcoded adapters (Bioo Scientific) were ligated to the DNA fragments which were then amplified for 13–18 PCR cycles. DNA fragment purification and size selection were performed using AMPure XP beads (Beckman Coulter). The libraries were sequenced by the Edinburgh Clinical Research Facility on Illumina NextSeq 550 or Illumina NextSeq 2000 instruments. Subsequent ChIP-seq data analysis was based on three replicates for CRD1 (BF), JBP2 (BF), and SET27 (PF), and two replicates for the other YFP-tagged proteins and for the untagged control in BF and PF cells. ChIP-seq data for CRD1 (BF), SET27 (BF), and the untagged control (BF) were taken from *Staneva et al., 2021*.

## ChIP-seq data analysis

ChIP-seq reads were first de-duplicated with pyFastqDuplicateRemover (*Webb et al., 2018*) and then aligned to the Tb427v9.2 genome (*Müller et al., 2018*) with Bowtie 2 (*Langmead and Salzberg, 2012*). Each ChIP sample was normalized to its input (as a ratio) and to library size (as counts per million). ChIP-seq data were visualized using IGV (*Robinson et al., 2011*). Metagene plots and heatmaps were generated as follows. First, CRD1 peaks were called using a combination of MACS2 (*Feng et al., 2012*) and manual filtering of false positives including peaks absent in some CRD1 replicates, peaks called in the untagged control cell line, and peaks with IP/input ratios <6.5. Then, 20 kb regions centered around each CRD1 summit were divided into 50 bp windows. Metagene plots were generated by adding the normalized reads in each 50 bp window and then representing them as a density around CRD1. The heatmaps show normalized reads around individual CRD1 peaks.

## RNA extraction, RT-qPCR, and RNA-seq

$1–5×10^7$ parasites were harvested by centrifugation and RNA was extracted from cells using TRIzol (Thermo Fisher Scientific) according to the manufacturer's protocol. RNA samples were treated with TURBO DNase (Thermo Fisher Scientific) followed by purification with Phenol/Chloroform/Isoamyl alcohol pH 4. RT-qPCR on extracted RNA was performed using Luna Universal One-Step RT-qPCR Kit (NEB). Primers used in the qPCR reactions are listed in *Supplementary file 4*. Libraries for RNA-seq were prepared and sequenced by the Edinburgh Clinical Research Facility using NEBNEXT Ultra II Directional RNA Library Prep kit (NEB) and Poly-A mRNA magnetic isolation module (NEB). The libraries were sequenced on Illumina NextSeq 550 or Illumina NextSeq 2000 instruments. Subsequent RNA-seq data analysis was based on two replicates for the wild-type, *jbp2Δ/+*, and *jbp2Δ/Δ* cell lines, and three replicates for the *set27Δ/+* and *set27Δ/Δ* cell lines.

## RNA-seq data analysis

RNA-seq reads were aligned to the Tb427v9.2 genome (*Müller et al., 2018*) with Bowtie 2 (*Langmead and Salzberg, 2012*). Aligned reads were separated into those originating from the plus or the minus DNA strand, normalized to library size (as counts per million) and visualized using IGV (*Robinson et al., 2011*). The differential expression analysis was performed as follows. First, raw non-normalized reads overlapping mRNAs annotated in the Tb427v9.2 genome were counted using featureCounts (*Liao et al., 2014*). Reads overlapping several mRNAs were assigned to the mRNA with the largest number of overlapping bases. Differential expression analysis was then performed using edgeR (*Robinson et al., 2010*) utilizing the TMM normalization method (*Robinson and Oshlack, 2010*). The filterByExpr (min.count=4; min.total.count=20) function was applied to filter out genes with insufficient counts for performing the statistical analysis.

## Affinity selections and mass spectrometry

$4×10^8$ cells were harvested by centrifugation and lysed in a buffer containing 50 mM Tris pH 8, 150 mM KCl, and 0.2% NP-40. Lysates were sonicated in a Bioruptor (Diagenode) sonicator for three cycles (each cycle: 12 s ON, 12 s OFF; high setting). Sonicated samples were centrifuged and the supernatant (soluble fraction) was incubated for 1 hr at 4°C with beads crosslinked to anti-GFP antibody (11814460001; Roche) or to M2 anti-FLAG antibody (F1804; Sigma-Aldrich). The beads were then washed three times with lysis buffer. Proteins were eluted from the beads using RapiGest surfactant (Waters) at 55°C for 15 min. Proteins were digested by filter-aided sample preparation (*Wiśniewski*

*et al., 2009*). Briefly, protein samples were reduced using DTT, then denatured with 8 M Urea in Vivakon spin column cartridges. Samples were treated with 0.05 M iodoacetamide and digested overnight with 0.5 µg MS Grade Pierce Trypsin Protease (Thermo Fisher Scientific). Digested samples were desalted using stage tips (*Rappsilber et al., 2007*) and resuspended in 0.1% trifluoroacetic acid for mass spectrometry analysis. LC-MS/MS was performed as described previously (*Staneva et al., 2021*). The results were processed using the Perseus software (*Tyanova et al., 2016*) and are based on three biological replicates per sample. Data for CRD1 (BF) and SET27 (BF) were taken from *Staneva et al., 2021*. The interaction network of the analyzed proteins in BF trypanosomes was created using Cytoscape (*Shannon et al., 2003*).

## Materials availability statement
Plasmids and cell lines made as part of this study are available upon request.

## Data access
The sequencing data generated in this study can be accessed on the NCBI Gene Expression Omnibus (GEO; https://www.ncbi.nlm.nih.gov/geo/) with accession number GSE208037. The proteomics data generated in this study can be accessed on the Proteomics Identification Database (PRIDE; https://www.ebi.ac.uk/pride/) with accession number PXD036454.

## Use of previously published data sets
We have used CRD1 (BF), SET27 (BF), and untagged (BF) ChIP-seq and proteomics data from *Staneva et al., 2021*. These can be accessed on NCBI GEO (https://www.ncbi.nlm.nih.gov/geo/) with accession number GSE150253 and on PRIDE (https://www.ebi.ac.uk/pride/) with accession number PXD026743.

## Acknowledgements
The authors thank Julie Young for supplying the HMI-9 media used during this project. The authors would also like to thank Richard Clark, Angie Fawkes, Audrey Coutts, and Tamara Gilchrist from the Edinburgh Wellcome Clinical Research Facility for sequencing services as well as Shaun Webb from the WCB bioinformatics core facility for maintaining the servers used for processing sequencing data. This work was funded by an MRC Research Grant awarded to R.C.A and K.R.M and supporting D.P.S (MR/T04702X/1), a Wellcome Investigator Award to K.R.M. (221717), a Wellcome Principal Research Fellowship to R.C.A. supporting D.P.S. and T.A (200885; 224358), a Wellcome Principal Research Fellowship to D.T. supporting S.B. (222516), a Wellcome Senior Research Fellowship to A.A.J. (202811), a Wellcome Instrument grant to J.R. (108504), and core funding for the Wellcome Centre for Cell Biology (203149) supporting C.S.

## Additional information

### Funding

| Funder | Grant reference number | Author |
| --- | --- | --- |
| Medical Research Council | MR/T04702X/1 | Desislava P Staneva<br>Keith R Matthews<br>Robin C Allshire |
| Wellcome Trust | 221717 | Keith R Matthews |
| Wellcome Trust | 200885 | Desislava P Staneva<br>Tatsiana Auchynnikava<br>Robin C Allshire |
| Wellcome Trust | 224358 | Desislava P Staneva<br>Tatsiana Auchynnikava<br>Robin C Allshire |
| Wellcome Trust | 222516 | Stefan Bresson<br>David Tollervey |

| Funder | Grant reference number | Author |
|---|---|---|
| Wellcome Trust | 202811 | A Arockia Jeyaprakash |
| Wellcome Trust | 108504 | Juri Rappsilber |
| Wellcome Trust | 203149 | Christos Spanos |

The funders had no role in study design, data collection and interpretation, or the decision to submit the work for publication. For the purpose of Open Access, the authors have applied a CC BY public copyright license to any Author Accepted Manuscript version arising from this submission.

## Author contributions

Desislava P Staneva, Conceptualization, Data curation, Formal analysis, Validation, Investigation, Visualization, Methodology, Writing – original draft, Writing – review and editing; Stefan Bresson, Formal analysis, Visualization, Writing – original draft, Writing – review and editing; Tatsiana Auchynnikava, Christos Spanos, Formal analysis, Methodology; Juri Rappsilber, Resources; A Arockia Jeyaprakash, Formal analysis; David Tollervey, Writing – review and editing; Keith R Matthews, Conceptualization, Resources, Supervision, Funding acquisition, Project administration, Writing – review and editing; Robin C Allshire, Conceptualization, Resources, Supervision, Funding acquisition, Writing – original draft, Project administration, Writing – review and editing

## Author ORCIDs

Desislava P Staneva (iD) http://orcid.org/0000-0003-1330-2158
Stefan Bresson (iD) http://orcid.org/0000-0002-0329-2243
Christos Spanos (iD) http://orcid.org/0000-0002-4376-8242
David Tollervey (iD) http://orcid.org/0000-0003-2894-2772
Keith R Matthews (iD) http://orcid.org/0000-0003-0309-9184
Robin C Allshire (iD) http://orcid.org/0000-0002-8005-3625

## Decision letter and Author response

Decision letter https://doi.org/10.7554/eLife.83135.sa1
Author response https://doi.org/10.7554/eLife.83135.sa2

# Additional files

## Supplementary files

• Supplementary file 1. Significantly enriched interactors detected by proteomic analysis of affinity selected YFP- and FLAG-tagged proteins. Cut-offs used to determine significant interacting partners: $\log_2(\text{tagged/untagged})>2$ and $P<0.01$ (Student's t-test).

• Supplementary file 2. Shared and unique interactors of affinity selected YFP-tagged proteins in BF *T. brucei*.

• Supplementary file 3. Differentially expressed mRNAs in *set27Δ/Δ* vs WT. Cut-offs used to determine significant hits: fold change (FC) >2 or <-2 and false discovery rate (FDR)<0.01.

• Supplementary file 4. Oligos used in this study. Upper case sequences align to *T. brucei* DNA. Lower case sequences align to pPOT plasmid DNA except for the sgRNA oligos targeting the *JBP2* gene (see the "Notes" column for these oligos). "F" denotes a forward primer and "R" denotes a reverse primer.

• MDAR checklist

## Data availability

The sequencing data generated in this study can be accessed on the NCBI Gene Expression Omnibus (GEO; https://www.ncbi.nlm.nih.gov/geo/) with accession number GSE208037. The proteomics data generated in this study can be accessed on the Proteomics Identification Database (PRIDE; https://www.ebi.ac.uk/pride/) with accession number PXD036454.

The following datasets were generated:

| Author(s) | Year | Dataset title | Dataset URL | Database and Identifier |
|---|---|---|---|---|
| Staneva DP, Bresson S, Auchynnikava T, Spanos C, Rappsilber J, Jeyaprakash AA, Tollervey D, Matthews KR, Allshire RC | 2022 | The SPARC complex defines RNAPII promoters in *Trypanosoma brucei* | https://www.ncbi.nlm.nih.gov/geo/query/acc.cgi?acc=GSE208037 | NCBI Gene Expression Omnibus, GSE208037 |
| Staneva DP, Bresson S, Auchynnikava T, Spanos C, Rappsilber J, Jeyaprakash AA, Tollervey D, Matthews KR, Allshire RC | 2022 | The SPARC complex defines RNAPII promoters in *Trypanosoma brucei* | https://www.ebi.ac.uk/pride/archive/projects/PXD036454 | PRIDE, PXD036454 |

The following previously published datasets were used:

| Author(s) | Year | Dataset title | Dataset URL | Database and Identifier |
|---|---|---|---|---|
| Staneva DP, Carloni R, Auchynnikava T, Tong P, Rappsilber J, Jeyaprakash AA, Matthews KR, Allshire RC | 2021 | A systematic analysis of *Trypanosoma brucei* chromatin factors identifies novel protein interaction networks associated with sites of transcription initiation and termination | https://www.ncbi.nlm.nih.gov/geo/query/acc.cgi?acc=GSE150253 | NCBI Gene Expression Omnibus, GSE150253 |
| Staneva DP, Carloni R, Auchynnikava T, Tong P, Rappsilber J, Jeyaprakash AA, Matthews KR, Allshire RC | 2021 | A systematic analysis of *Trypanosoma brucei* chromatin factors identifies novel protein interaction networks associated with sites of transcription initiation and termination | https://www.ebi.ac.uk/pride/archive/projects/PXD026743 | PRIDE, PXD026743 |

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
