## [Editor Report]

Trypanosomes are unusual in the way that they transcribe protein-coding genes. This work defines the role of the SPARC complex in transcription and highlights the role of potential histone readers and writers. This work will be of interest to those in the kinetoplastid community, especially those working on the control of gene expression.

---

## [Decision Letter]

[Editors' note: this paper was reviewed by Review Commons.]

---

## [Author Response]

Reviewer #1 (Evidence, reproducibility and clarity (Required)):In this paper, Staneva et al. describe a novel complex found at RNA PolII promoters that they term the SPARC. The manuscript focuses on defining the core components of the complex and the pivotal role of SET27 in defining its function, and role in PolII transcription. This manuscript is a logical follow on from an initial paper (Staneva et al., 2021) by the same authors where they systematically analyzed chromatin factors, and their role in both transcription start and termination. What is also very clear, is that this complex is one made of histone readers and writers which suggests its function is to change the chromatin structure around a PolII promoters. The authors show that this complex is necessary for the correct positioning of PolII and directionality of transcription.This was a well-designed study and well written and clear manuscript that provides fascinating insight transcription control in bloodstream form parasites.I have no major comments only a few minor ones.1) Localisation of the different SPARC components appears to be either nuclear or nuclear and cytoplasmic.– Both SET27 and CRD1 show a nuclear and cytoplasmic localisation in the bloodstream form IFA (Supplementary Figure 1B), but only a nuclear localisation procyclic form.Did the authors attempt C terminally tagging SET27, CRD1 to see if this resulted in a change in the pattern?

We have not tagged either protein at the C terminus, however SET27 (Tb927.9.13470) has been tagged both N- and C-terminally in procyclic form (PF) cells as part of the TrypTag project (http://tryptag.org). In both cases, SET27 localized to the nucleus, suggesting that the differences in localization we observe for SET27 depend on the life cycle stage, and not on the position of the tag. One caveat is that in the TrypTag project proteins are tagged with mNeonGreen whereas in our study proteins were tagged with YFP. Based on our images, CRD1 appears to be predominantly nuclear in both bloodstream form (BF) and PF parasites. CRD1 (Tb927.7.4540) has been tagged only N-terminally in PF cells as part of the TrypTag project where it has also been classified as mostly nuclear with only 10% of cells showing cytoplasmic localization for CRD1.

We are well aware that tags can alter the behaviour of a protein. Absolute confirmation of location will require the generation of antibodies that detect untagged proteins. However, this is a longer-term undertaking. We have added the following statement to the Results section to address the point raised:

“We tagged the proteins on their N termini to preserve 3′ UTR sequences involved in regulating mRNA stability (Clayton, 2019). We note, however, that the presence of the YFP tag and/or its position (N- or C-terminal) might affect protein expression and localization patterns”.

– The point is made that JBP2 shows a 'distinct cytoplasmic localisation' in PF cells. by this logic, the SET27 localisation in BF is also distinctly cytoplasmic and a nuclear enrichment is not clear.

Indeed the reviewer is correct – we have inadvertently over accentuated the significance of this difference in the text. We had emphasized the predominantly cytoplasmic localization of JBP2 in PF trypanosomes as potentially related to its weaker association with other (predominantly nuclear) SPARC components in the mass spectrometry experiments. The presence of SET27 in the nuclei of both BF and PF cells is confirmed by a positive ChIP signal. We have revised the manuscript text by changing “distinct cytoplasmic” to “predominantly cytoplasmic” to describe JBP2 localization in PF cells. We hope that this resolves the issue.

– Why would the localisation pattern change between life cycle stages? Surely PolII transcription should remain the same?

Although our analysis suggests that there may be some shift in SET27 and JBP2 localization between BF and PF stages, sufficient amounts of these proteins may be present in the nucleus for proper SPARC assembly and RNAPII transcription regulation in both life cycle forms. The proportion of SET27 and JBP2 proteins that localizes to the cytoplasm may have functions unrelated to transcription.

2) Several of the images in Supplementary Figure 1B seem to show foci in the nucleus (CSD1, PWWP1, CRD1). Do you see foci throughout the cell cycle or just in G1/S phase cells as shown here?

We have not systematically investigated protein localization at different cell cycle stages, so we do not have microscopy images for all proteins at all stages of the cell cycle. However, the images we did collect suggest the punctate pattern is preserved for CRD1 in the G2 phase in both BF and PF cells (Author response image 1) as we showed in Figure 1 —figure supplement 1B for cells with 1 kinetoplast and 1 nucleus (G1/S phase cells). The significance of these puncta remains to be determined.

**Author response image 1. sa2fig1:** 

3) In Figure 6, what does 'TE' stand for?

TE denotes transposable elements. We have added this to the figure legend.

4) The authors show this interesting link between SPARC complex and subtelomeric VSG gene silencing.– In the CRD1 ChIP or RBP1 ChIP, are there any other peaks in telomere adjacent regions in the WT cells similar to that seen on chromosome 9A? And does the sequence at this point resemble a PolII promoter?

Apart from peaks located on Chromosome 9_3A, there are other CRD1 and RPB1 ChIP peaks in chromosomal regions adjacent to telomeres in WT cells. We observed broadening of RPB1 distribution in these regions upon SET27 deletion, similar to what we show for Chromosome 9_3A. In particular, wider RPB1 distribution on Chromosome 8_5A coincides with upregulation of 10 VSG transcripts. These two loci explain most of the differentially expessed genes (DEGs) detected, but other subtelomeric regions show a similar pattern. We have added the following statement to the Results section to highlight that the phenotype shown for Chromosome 9_3A is not unique:

“We also observed a similar phenotype at other subtelomeric regions, such as Chromosome 8_5A where 10 VSGs and a gene encoding a hypothetical protein were upregulated upon SET27 deletion (Supplementary file 3)”.

Cordon-Obras et al. (2022) have recently defined key sequence elements present at one RNAPII promoter. We searched for similar sequence motifs but failed to identify them as underlying CRD1 and RPB1 ChIP peaks, highlighting the likely sequence heterogeneity amongst trypanosome RNAPII promoters. To address this point, we have added the following sentence to the Discussion:

“Sequence-specific elements have recently been found to drive RNAPII transcription from a *T. brucei* promoter (Cordon-Obras et al., 2022), however, we were unable to identify similar motifs underlying CRD1 or RPB1 ChIP-seq peaks, suggesting that *T. brucei* promoters are perhaps heterogeneous in composition”.

– In the FLAG-CRD1 IP (Figure 3B), the VSG's seen here are not represented (as far as I can tell) in Figure 6B and C. If my reading is correct could, is this a difference in the FC cut off for what is significant in these experiments?

The VSGs detected in the FLAG-CRD1 IP from *set27*D/D cells are indeed different from the ones shown in Figure 6 (even after setting the same fold change cutoffs). We have highlighted this by adding the following statement to the Results section: “Gene ontology analysis of the upregulated mRNA set revealed strong enrichment for normally silent VSG genes (Figure 6B-D) which were distinct from the VSG proteins detected in the FLAG-CRD1 immunoprecipitations from *set27*D/D cells (Figure 3B)”.

The VSGs in the mass spectrometry experiments likely represent unspecific interactors of FLAG-CRD1. To clarify this, we have added the following statement to the Results section: ”Instead, several VSG proteins were detected as being associated with FLAG-CRD1 in *set27*D/D cells, though it is likely that these represent unspecific interactions”.

Reviewer #1 (Significance (Required)):Trypanosomes are unusual in the way that they transcribe protein coding genes. Recent advances have defined the chromatin composition at the TSS and TTS, and the recent publication of a PolII promoter sequence(s) further adds to our understanding of how transcription here is regulated. Defining the SPARC complex now add to this understanding and highlights the role of potential histone readers and writers. I think that this will be of interest to the kinetoplastid community especially those working on control of gene expression.Our lab studies gene expression and antigenic variation in *T. brucei*.Reviewer #2 (Evidence, reproducibility and clarity (Required)):In this manuscript, the authors identify a six-membered chromatin-associated protein complex termed SPARC that localizes to Transcription Start Regions (TSRs) and co-localizes with and (directly or indirectly) interacts with RNA polymerase II subunits. Careful deletion studies of one of its components, SET27, convincingly show the functional importance of this complex for the genomic localization, accuracy, and directionality of transcription initiation. Overall, the experiments are well and logically designed and executed, the results are well presented, and the manuscript is easy to read.There are a few minor points that would benefit from clarification and/or from a more detailed discussion:1) The concomitant expression of many VSGs (37) in a SET27 deletion strain is remarkable and has important implications for their normally monoallelic expression. It is well established that VSG expression in wild-type *T. brucei* can only occur from one of ~15 subtelomeric bloodstream expression sites, which include the ESAGs. This result implies that VSG genes are also transcribed from "archival VSG sites" in the genome, not only from expression sites. Are there VSGs from the silent BESs among the upregulated VSGs? Is there precedence in the literature for the expression of VSGs from chromosomal regions besides the subtelomeric expression sites?

Our analysis of differentially expressed genes (DEGs) revealed that 43 VSG genes (37 of which are subtelomeric) and 2 ESAG genes are upregulated in the absence of SET27. Both ESAGs but none of the upregulated VSGs in *set27*D/D cells are annotated as located in BES regions. While it is possible that recombination events have resulted in gene rearrangements between the reference strain and our laboratory’s strain, at least some of the upregulated VSGs are likely to be transcribed from non-BES archival sites. VSG transcript upregulation from non-BES regions was also recently described by López-Escobar et al. (2022).

We note that the upregulated mRNAs in *set27*D/D are still relatively lowly expressed (Figure 6C). This is presumably insufficient to coat the surface of *T. brucei*, and expression from BES sites instead may be required to achieve this. We have revised the manuscript Discussion section to make these points more clear:

“Bloodstream form trypanosomes normally express only a single VSG gene from 1 of ~15 telomere-adjacent bloodstream expression sites (BESs). In contrast, in *set27*D/D cells we detected upregulation of 43 VSG transcripts, none of which were annotated as located in BES regions. Recently, López-Escobar et al. (2022) have also observed VSG mRNA upregulation from non-BES locations, suggesting that VSGs might sometimes be transcribed from other regions of the genome. However, the VSG transcripts we detect as upregulated in *set27*D/D were relatively lowly expressed (Figure 6C) and may not be translated to protein or be translated at low levels compared to a VSG transcribed from a BES site”.

2) The role of SPARC in defining transcription initiation is compelling. It's less clear to the reviewer if the observed transcriptional silencing within subtelomeric regions can also ascribed to SPARC. Have the authors considered the possibility that some components of the SPARC may be shared by other chromatin complexes, which could be responsible for the transcriptional activation of silent genes in SET27 deletion mutants?

We cannot rule out indirect effects through the participation of some SPARC components in other complexes operating independently of SPARC. Indeed, the transcriptional defect within the main body of chromosomes appears to be somewhat different from that observed at subtelomeric regions, particularly with respect to distance from SPARC. We have added a statement in the Discussion section to highlight the possibility raised by the reviewer:

“However, an alternative possibility is that transcriptional repression in subtelomeric regions is mediated by different protein complexes which share some of their subunits with SPARC, or whose activity is influenced by it”.

3) The authors mention that the observed interaction of FLAG-CRD1 with VSGs in the immunoprecipitations (Figure 3B) is evidence for the actual expression of normally silent VSGs on the protein level. This is true, but it should be spelled out that this interaction is nevertheless likely an artifact, at least the physiological relevance of these interactions is questionable.

We agree that these are likely background associations and have added the following statement to the Results section to clarify this point:

“Instead, several VSG proteins were detected as associated with FLAG-CRD1 in *set27*D/D cells, though it is likely that these represent unspecific interactions”.

To avoid unnecessary confusion we have also removed the following sentence from the revised Discussion:

“The interactions of FLAG-CRD1 with VSGs in the affinity selections from *set27*Δ/Δ cells indicate that some of the normally silent VSG genes are also translated into proteins in the absence of SET27”.

4) "ophistokont" is misspelled in the introduction

Thanks for noticing. We have corrected it to “Opisthokonta”.

Reviewer #2 (Significance (Required)):The manuscript by Staneva et al. addresses the fundamental regulatory mechanism of gene transcription in the protozoan parasite *Trypanosoma brucei*, a highly divergent eukaryotic organism that is renowned for unusual features and mechanisms in gene regulation, metabolism, and other cellular processes. While post-transcriptional regulation is prevalent and relatively well established in T. brucei, much less is known about the mechanism of transcription initiation and transcriptional control, in part due to the general paucity of well-defined conventional promoter regions in this organism (only very few have been identified thus far). In this context, the work by Staneva et al. is highly significant and represents an important contribution to the field of gene regulation and chromatin biology in *T. brucei* and other related kinetoplastid parasites.